Journal of
open psychology data

# Data From a Validation Study of Two Psychometric Models on Test-taking Behavior

DATA PAPER

SÖREN MUCH** 🆔

AUGUSTIN MUTAK** 🆔

STEFFI POHL 🆔

JOCHEN RANGER 🆔

*Author affiliations can be found in the back matter of this article

**Shared first authorship

]u[ ubiquity press

## ABSTRACT

This paper introduces a dataset from a validation study of two psychometric models, one on the intraindividual speed-ability relationship and the other one on persistence. It includes responses, response times, and action sequences from $N = 1244$ participants who completed a matrix reasoning test under two experimental conditions, one being speeded and one being non-speeded. Additionally, it includes measures on motivational disposition, current motivation, effort, and concentration. Collected online via Prolific, the data is freely available at OSF (https://osf.io/9j6hm/). This dataset may aid in the development and validation of psychometric models on response processes as well as the investigation of test-taking behavior.

CORRESPONDING AUTHOR:
**Sören Much**

Martin-Luther-Universität
Halle-Wittenberg, DE;
Freie Universität Berlin, DE

soeren.much@fu-berlin.de

KEYWORDS:
test-taking behavior;
psychometrics; persistence;
speed-ability trade-off

TO CITE THIS ARTICLE:

# (1) BACKGROUND

It has been consistently shown that test results in low-stakes assessments are affected by test-taking behavior. Not accounting for such behavior may heavily distort validity and fairness of the results (e.g., Pohl et al., 2021; Singer & Braun, 2018). Drawing on response and response time data, a lot of psychometrics research dealt with modeling and investigating test-taking behavior, such as guessing (e.g., Guo et al., 2016; Schnipke & Scrams, 1997; Wang & Xu, 2015), missing values (Weeks, von Davier, & Yamamoto, 2016; Pohl et al., 2019), disengagement (e.g., Pokropek, 2016; Ulitzsch et al., 2020) or the intra-individual relation of ability and speed (e.g. Domingue et al., 2022; Kang et al, 2022; Ranger et al., 2021). The dataset presented in this paper was collected to validate two recently developed psychometric models on test-taking behavior. The first model aims at modeling the intraindividual speed-ability relationship and the second one aims to model persistence in a test.

The Individual Speed-Ability Relationship (ISAR) model (Mutak et al., 2024) is based on van der Linden's (2007) hierarchical speed-accuracy (SA) model. It models the latent ability ($\theta$) via a conventional IRT model (e.g. Rasch or 2PL) based on item responses. Latent speed ($\tau$) of the respondents is modeled via a log-normal model based on item-wise response time data. A linear latent growth term is specified for both ability and speed, capturing the intra-individual change in these constructs across the test. Relying on non-stationarity of ability and speed, we may observe a variety of ability and speed levels within a person, and as such investigate their intra-individual relationship. The correlation between the slope in latent ability and the slope in latent speed may be used to examine the speed-ability trade-off (SAT). However, other confounding aspects such as change in motivation or concentration may impact both types of changes, and as such hinder the interpretation of the correlation as representing the SAT. With the present dataset we aimed to validate the interpretation of this parameter as a measure of the SAT. We also aimed to investigate to which extent concentration and motivation may confound this relation and to control for these confounding variables.

The Linear Ballistic Accumulator Model for Persistence (LBA-P; Ranger et al., 2024 [submitted]) stems from the tradition of sequential sampling models and extends the Linear Ballistic Accumulator model by Brown and Heathcote (2008) by a persistence component. The LBA-P describes the process of solving an item as a race of accumulators that are distinctly determined by the ability of test-takers and their persistence. It assumes a persistence accumulator that governs when a test-taker stops working on an item, a second accumulator that represents the generation of a correct response, and a third accumulator that represents the generation of a wrong response. The accumulator that hits the response threshold first determines the response, with the response time being the hitting time. When the persistence accumulator hits first, the response is determined by the last status of the other accumulators: In case both accumulators are below a second threshold, the omission threshold, the item is omitted. This assumes that test-takers do not respond when they have little information on the response. When at least one of the other accumulators surpassed the omission threshold, the model assumes that the test-taker makes an informed guess: The test-takers chose the response corresponding to the accumulator with the higher value. Each accumulation process starts equally at zero and is characterized by its drift rate. The drift rates are governed by the latent traits $\theta_1$, (representing disengagement/negative persistence), $\theta_2$ (ability), and $\theta_3$ (error-proneness). The presented data are collected to validate the interpretation of the $\theta_1$ parameter as a measure of persistence.

The preregistration for the validation study of both models (Mutak & Much et al., 2023) is available in the online repository at https://osf.io/jm7ne.

# (2) METHODS

## 2.1 STUDY DESIGN
### Experimental design
The design of the study is depicted in Figure 1. The study focused on the responses and response time data from a matrix reasoning test, to which the models to be validated

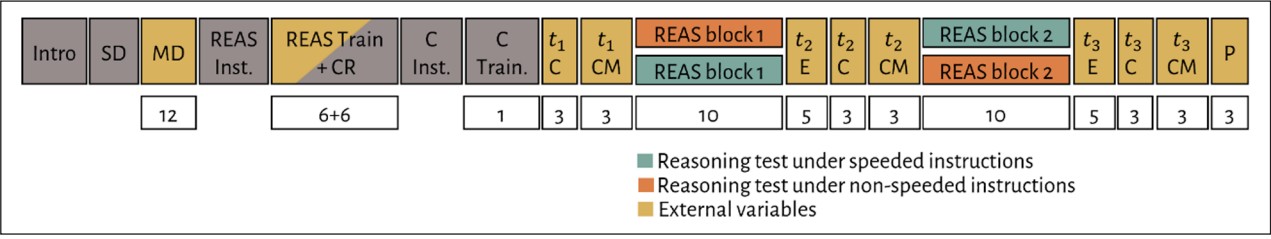

**Figure 1** Design of the Study.

*Note.* The numbers below the boxes describe the number of items used for each measure. Abbreviations: Intro = Introduction, SD = Sociodemographic information, MD = Motivational Disposition, REAS = Reasoning test, Inst. = Instructions, Train. = Training, CR = Confidence ratings, C = Concentration, CM = Current motivation, E = Effort, P = Persistence.

will be applied. This study used an experimental design with two factors, one being varied between groups and one within groups. As a within-group factor we varied instructions on how to approach the test: one was a non-speeded condition, in which speed was de-emphasized; the other was a speeded condition, in which speed was emphasized.[1] At the between-subject level the order of the instructions was varied, the first group starting with the non-speeded instruction, and the second group with the speeded one. Participants were assigned randomly to a group. The assignment was conducted directly at the beginning of the study.

Participants were partially blind towards the manipulation. As the manipulation consists of instructing the participants to work at different speed levels in blocks of the test, the participants were not blind to the within-subject manipulation. However, they were blind to the fact that there is another group with a different instruction order. Furthermore, participants were not aware that test-taking behavior is being investigated. At the beginning of the study, they were informed that the study aim is to investigate a newly developed test for logical reasoning. A debriefing was done at the end of the study. Participants could withdraw their consent to use their data after debriefing.

### Procedure

The survey started with introductory information about the study, declaration of consent, and information on data usage and the privacy policy. This introduction was followed by socio-demographic questions (SD) and the assessment of motivational disposition (MD). Participants then received the instructions on how to approach the reasoning test (REAS Inst.) in the form of a tutorial to acquaint themselves with the task format and the way responses are given. After that, they received six training items (REAS Train.) that were combined with confidence ratings on their performance after each training item (CR). After the reasoning training block, participants received instructions and training for the concentration task. The participants took the matrix-reasoning test in two 10-item blocks (REAS block 1, REAS block 2), each block under a different instruction. Before and after each of these two blocks, concentration ($t_1$ C, $t_2$ C, $t_3$ C) and current motivation ($t_1$ CM, $t_2$ CM, $t_3$ CM) were assessed. After each of the two test blocks, test-taking effort ($t_2$ E and $t_3$ E) was assessed retrospectively. The participants were encouraged to take short breaks after each effort assessment. Finally, we assessed persistence (P) as an external measure. At the end of the study, participants were debriefed about the study aim and were given the opportunity to revoke their consent.

The study was implemented with the online survey platform SoSci Survey. The median study duration was 50.48 minutes (*SD* = 24.88).

## 2.2 TIME OF DATA COLLECTION
November 2023–January 2024

## 2.3 LOCATION OF DATA COLLECTION
Data was collected worldwide using the online study recruitment platform Prolific.

## 2.4 SAMPLING, SAMPLE AND DATA COLLECTION
### 2.4.1. Sampling
Participants were recruited via the online platform Prolific. They were paid £7,50 for participation. To be included in the study, participants had to fulfil the following criteria:

- fluency in English,
- age between 18 and 40 years,
- access the study via a tablet or a computer (no mobile phone) to ensure that items are displayed correctly,
- an approval rate on the platform of >99%, and,
- no dyslexia (medically diagnosed, in the process of being diagnosed or strongly suspecting and undiagnosed dyslexia), as the concentration task includes the distinction of letters.

In the following cases the participants' data were excluded from the dataset and the participant received no compensation for their participation: a) The participant has failed all three attention checks, b) The participant's total log response time on the test is less than 3 standard deviations below the mean, and c) The last page of the study wasn't reached.[2]

The target sample size was 1000 cases,[3] with 800 cases randomly assigned to the first experimental group and 200 cases to the second.[4]

### 2.4.2. Sample description
We collected finished trials from 1274 paid participants who passed at least one attention check. From this total sample, we excluded 4 participants (4 from experimental group 1 and 0 from experimental group 2) that revoked their consent to use the data after debriefing them about the study goals. Further, we excluded 9 participants who did not match the dyslexia criterion (7 from experimental group 1 and 2 from experimental group 2), and 17 participants who restarted the study after they already had begun the reasoning training in a previous attempt (16 from experimental group 1 and 1 from experimental group 2).

The actual sample size turned out to be *N* = 1244 participants, with 992 being assigned to the first experimental group, and 252 to the second. The sample included 717 persons self-identifying as male, 512 persons self-identifying as female and 15 persons responding with "other". The mean age in the sample was 28 years (*SD* = 5.48). The education level in the sample is shown in Table 1. The native language and

| EDUCATION LEVEL | FREQUENCY |
|---|---|
| Still in school | 31 (2.5%) |
| Finished school with no formal qualifications | 9 (0.7%) |
| Secondary school | 45 (3.6%) |
| High school/A-Levels | 215 (17.3%) |
| Technical/Community college | 106 (8.5%) |
| Undergraduate degree | 514 (41.3%) |
| Graduate degree | 288 (23.2%) |
| Doctorate | 30 (2.4%) |
| Other school-leaving qualification | 6 (0.5%) |

**Table 1** Frequency table on highest level of education.

| COUNTRY | FREQUENCY |
|---|---|
| South Africa | 248 (19.9%) |
| United Kingdom | 225 (18.1%) |
| Poland | 123 (9.9%) |
| United States | 110 (8.8%) |
| Portugal | 95 (7.6%) |
| Canada | 58 (4.7%) |
| Italy | 51 (4.1%) |
| Germany | 48 (3.9%) |
| Hungary | 38 (3.1%) |
| Mexico | 37 (3.0%) |
| Greece | 31 (2.5%) |
| Other (N < 30) | 180 (14.5%) |

**Table 2** Frequency table on country of current residence.

| LANGUAGE | FREQUENCY |
|---|---|
| English | 622 (50.0%) |
| Polish | 116 (9.3%) |
| Portuguese | 94 (7.6%) |
| Spanish | 73 (5.9%) |
| Italian | 51 (4.1%) |
| Hungarian | 32 (2.6%) |
| Other (N < 30) | 256 (20.6%) |

**Table 3** Frequency table on native language.

country of current residence is shown in Tables 2 and 3, respectively. 1234 participants solved the survey via a desktop computer, and 10 via a tablet.

## 2.5 MATERIALS/SURVEY INSTRUMENTS

### 2.5.1 Sociodemographic data

We assessed participants' gender ("male", "female", or "other"), age in years, native language ("English" or open input for other) and country of current residence (selection of ISO 3166-1 country as suggested after text input). The highest completed level of education was to be chosen from ("still in school", "Finished school with no formal qualifications", "Secondary school (e.g. GED/GSCE)", "High school diploma/A-levels", "Technical/community college", "Undergraduate degree (BA/BSc/other)", "Graduate degree (MA/MSc/MPhil/Diploma/Magister/other)", "Doctorate (PhD/Dr./other)", or "Other school-leaving qualification" with open input on the latter.

The last question in this block was whether they received a medical diagnosis for dyslexia that could be answered with "No", "No, but I strongly suspect I have undiagnosed dyslexia", "No, but I am in the process of being diagnosed", "Yes, I have been medically diagnosed with dyslexia", or "Rather not say".

The participants were pre-screened on Prolific for the inclusion criteria. If participants did not meet the criteria according to their responses in the study, they were redirected to Prolific directly after the sociodemographic questions and asked to return their submission.

### 2.5.2 Reasoning

To measure reasoning, we selected 20 items from the Open Matrices Item Bank (OMIB; Koch et al., 2022). An example item is depicted in Figure 2. Unlike other matrix tests, test-takers are supposed to construct their response by selecting a subset out of 20 figural elements to form the correct solution. A correct solution requires the row-wise application of 1 to 5 rules to infer the correct elements of the empty cell of the matrix. In the example in Figure 2, two rules are applied: addition (black corner triangles) and subtraction (sloped lines). The correct solution is formed by selecting elements 1 and 4 of the black corner triangles in the first row of the response options, as well as elements 1 and 3 of the sloped lines in the second row.

From the item bank containing 220 figural matrix items, 10 item pairs with an identical rule base and similar difficulty and time intensity were selected. From these item pairs, we assembled two parallel item sets with similar difficulty range and total test time intensity, each set forming one block of items. The items were presented in a fixed order for all participants. The item bank is available online at https://osf.io/4km79/. An overview of the selected items, including the pictured matrices, solutions and their rule base is available in the study's repository at https://osf.io/r43zf (direct document link).

The participants received an interactive task instruction containing two examples. Correct elements for the examples were highlighted. Participants could only proceed after selecting the correct subset of elements. After the interactive task construction, participants received six training items. The six training items covered all possible rules and all possible numbers of rules within an item to cover all difficulty levels. The start of each of the two main blocks was announced on a separate page with a reminder that revisiting items is not possible and

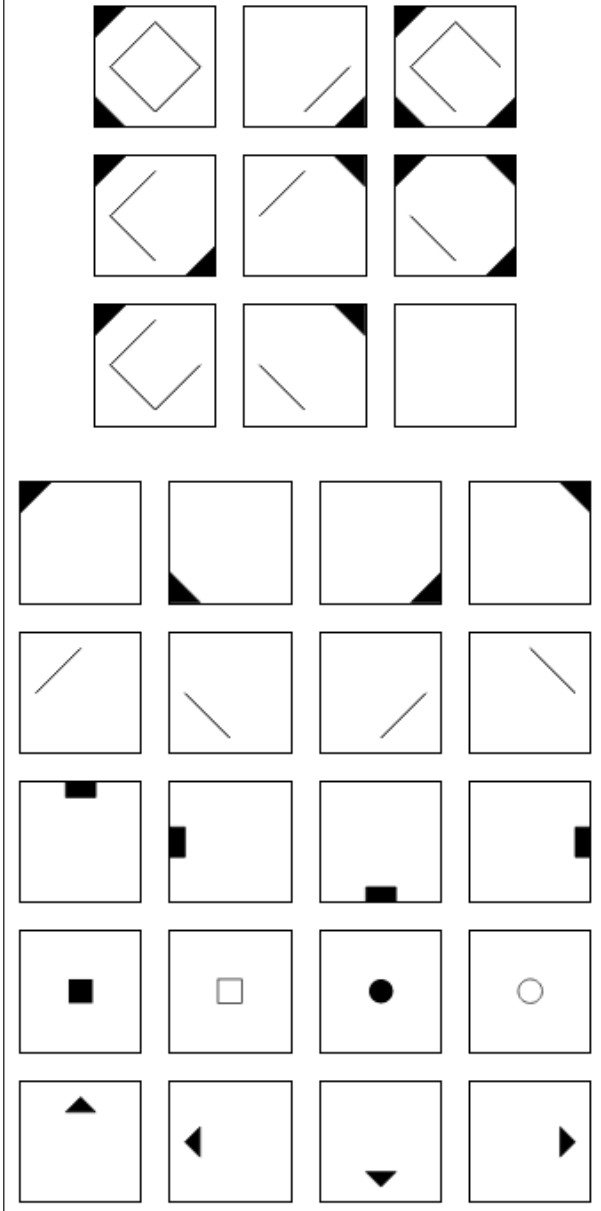

**Figure 2** Example of an item from the Open Matrices Item Bank (Koch et al., 2022). In the upper part, the 3 × 3-matrix task is presented, with the empty target element in the bottom right. In the lower part, the participant is asked to select all elements that form the correct solution (elements 1 and 4 from line 1 [addition] and elements 1 and 3 from line 2 [subtraction]). The item bank is available online at https://osf.io/4km79/.

with a highlighted text box containing the block-specific instructions to focus on either accuracy or speed. During the test, the current instruction was displayed at the top right of the page.

The selection and deselection of response elements was recorded with the corresponding times. We derived both accuracy and response times for each item from this. Participants were not required to select any element to get to the next item, so omissions were allowed.

### 2.5.3 Concentration

To measure current concentration, we devised a letter-counting task. Such measures are commonly used to

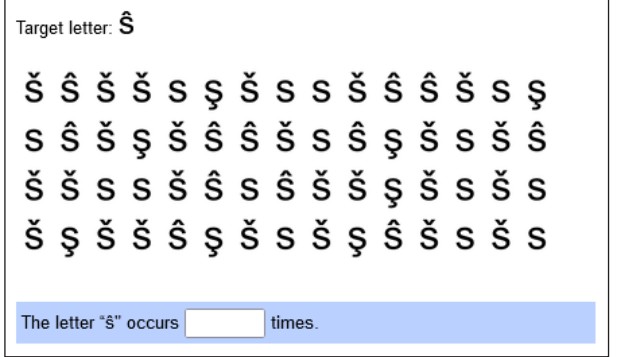

**Figure 3** Example of a concentration task item. The top line includes the target letter which is meant to be counted in the letter grid below. The response is to be typed into text field below.

assess current concentration (e.g., Brickenkamp et al., 2010). A letter counting task consists of several rows of visually similar letters, e.g., o, ó, ò, and ö. One of the letters is the target letter and in the original test the task of the participant is to cross out all target letters in the test within a given time limit. As this is not easily applicable to computerized testing, we adapted the approach. An example of our adaptation is depicted in Figure 3. In our adaptation, within each item, letters were depicted in 4 rows with 15 letters in each row. After a fixation cross at the center of the grid was displayed for 2 seconds, participants saw the letters for 20 seconds after which the grid disappeared. Participants were prompted to enter the number of target letters they counted in a text box that was visible during the whole task item. We used three different letter sets in these tasks and constructed three tasks for each letter set aiming for a similar difficulty within each letter set. In the study, at each time point concentration was measured with three task items (one for each letter set). All concentration task items are available online in the study's repository at https://osf.io/htrmu (direct document link). The response to each concentration task is scored either as being correct or incorrect. A concentration score is calculated for each person and each time point as the sum of the three task scores.

### 2.5.4 Motivational disposition

To assess motivational disposition, we used the 12-item subscale "Action orientation subsequent to failure vs. preoccupation" (AOF) from the Action Control Scale (ACS; Kuhl, 1994). This scale measures self-reported action vs. state orientation as an aspect of motivational disposition. There are two response options for each item, indicating either an action or a state orientation. Participants cannot omit responses to these items. Following the original test (Kuhl, 1994), scale scores are obtained by computing the sum score across the responses of all items for each participant. Due to the copyright of the material, it cannot be republished in the study's repository but is available from the cited source.

### 2.5.5 Persistence

For assessing persistence, we constructed two unsolvable items in the OMIB format, each by combining one rule from the regular rule set with two "non-rules". Within the non-rules, the progression of the given matrix cells is contradictory, e.g. an addition in the first row in combination with subtraction for the same group of elements in the second row or the faulty application of a rotation within one row. To mask that the items are unsolvable, these were preceded by a regularly solvable, but difficult item from the OMIB item bank. The pictured matrices and their rule bases are available online in the study's repository at https://osf.io/r43zf (direct document link). Again, participants were not required to select any response element to get to the next item, so omissions were allowed. As a measure of persistence, we assessed the time participants kept working on each of the two unsolvable items.

### 2.5.6 Test-taking effort

Test-taking effort was assessed using the 5-item "Effort" subscale from the Test-taking Motivation Instrument (TTMI) by Knekta and Eklöf (2015). Item phrasing was changed to refer to the last block of items instead of a test or a school test in general. The original instrument is available from Knekta and Eklöf (2015), the rephrased items are available online in the study's repository at https://osf.io/hnr93 (direct document link). All items were answered on a 4-point Likert-type scale ranging from 1 (strongly disagree) to 4 (strongly agree). Participants could not omit responses to these items. A scale score was computed for each participant as the mean of all his/her item responses.

### 2.5.7 Current Motivation

To assess current motivation, we rephrased 3 items of the 5-item "Effort" subscale from the Test-taking Motivation Instrument (TTMI) by Knekta and Eklöf (2015), so that they refer to the upcoming item block to measure participants' current willingness to put effort into solving the next block of items instead of tests in general. For the remaining two items, the rephrasing would be semantically unsuitable. The original instrument is available from Knekta and Eklöf (2015), the rephrased items are available online in the study's repository at https://osf.io/hnr93 (direct document link). All items were answered on a visual analogue scale from strongly disagree to strongly agree with an internal range of 101 points. The selected numerical value was not displayed. Participants could not omit responses to these items. A scale score was computed for each participant as the mean of all his/her item responses.

### 2.5.8 Test-taking behavior

For quality control and to assess qualitative variables on test-taking behavior, at the end of the questionnaire, participants were asked about their experience and their test-taking behavior within the study. First, participants were asked to indicate the number of items they guessed randomly and the number of items they guessed based on a partial solution. The responses were given on sliders with fixed markings at all numbers from 0 to 20. After that, participants were asked three questions in an open response format: "If you have skipped one or multiple items of the logical reasoning test, what were the reasons?", "Were there any issues (disruptions, confusing parts of the study etc.) that might have impacted your performance?", "Is there something you want to tell us regarding your experience with the study?", and "Do you have suggestions for improvements?"

## 2.6 QUALITY CONTROL

To ensure the quality of the study procedure and the resulting data, we conducted a calibration study, a pre-test, and established quality controls during the data collection. Furthermore, the application of exclusion criteria described in the sampling procedure (section 2.4.1) enhances the quality of the dataset.

### 2.6.1 Calibration study

We conducted a calibration study in order (a) to pre-test the selected reasoning items, the unsolvable items, as well as the concentration tasks and (b) to obtain item parameters of the matrix reasoning test items for linking purposes. In the calibration study we used a prototype of the self-developed concentration task, two blocks of matrix reasoning items in a non-speeded condition, and two self-developed unsolvable reasoning items. 200 students from two German universities took part in the calibration study. Participants could choose to receive instruction in German or English.

The calibrated item parameters for the matrix reasoning items were obtained using van der Linden's (2007) SA model. The parameter values are available in Table S1 of the validation study preregistration (Mutak & Much et al., 2023). 2PL model parameters of the OMIB items can be obtained from Table S1 in Koch et al. (2022) and the OMIB repository at https://osf.io/4km79/.

Concentration tasks were displayed without a time limit and included five different target letters. We evaluated the number of correct answers as well as the response time for each item. The results were used to (i) select the concentration tasks to be used in the main study and (ii) to set the time limit for the concentration task. We chose a time limit of 20 seconds, as this created enough variance in the responses without causing excess frustration in participants.

The unsolvable matrix items to assess persistence were included at the end of the calibration study procedure. Response times on these items were on average similar or larger than the response times to the most time-intensive regular items. This indicated that participants tried to solve these items with similar endurance as in the regular test and hence confirmed the basic functionality of the unsolvable items.

### 2.6.2 Pretest of the procedure

To estimate the required completion time of the study and to verify the functionality of the procedure, we conducted a pretest of the full study protocol (see section 2.1) with 17 German-speaking first-year psychology students in a German university (self-identified gender: 4 male, 13 female, no other). Their mean age was 20 years, ranging from 17 to 28. The study was conducted via the regular website, but in a group setting in a computer lab. Open-ended feedback by the participants and colleagues who tested the study led to improvements of instructions and the visual design of the study. As a couple of participants mentioned that they had a strong suspicion that the unsolvable items were intentionally not solvable, we added a regular, difficult item as the lead-in to the last block to maintain believability of the unsolvable items.

### 2.6.3 Exclusion criteria and attention checks

By conducting the study on Prolific, we relied on the platform's quality control and their measures to maintain a pool of verified and reliable participants. We included only participants with an approval rate of >99% on the platform.

During the data collection, we included three attention checks that were integrated at the three timepoints of the current motivation scale. The checks are instructional manipulation checks that instruct participants to select a certain response. They are displayed in the same response format as the accompanying items with a visual analogue scale. Participants that failed all three attention checks were excluded from the study. An attention check was considered as failed when the response on the 100-point visual analogue scale was more than 20 points off the response participants were instructed to select. The number of passed attention checks is included in the dataset as variable *AC*.

### 2.6.4 Qualitative variables of test-taking behavior

From the open responses to questions of test-taking behavior and the subjects' experiences, we formed a nominal variable *Disruptions* to subsume and categorize participants' responses to these open questions. This included disruptions during the test, the occurrence of technical difficulties, indications of a lack of understanding of the presented tasks, or discomfort. Participants reported these issues in all of the three open questions, even though it was only expected in the question on disruptions. Therefore, we used the responses given in all questions on test-taking behavior to form the nominal variable. Based on a review of all responses, we chose the following 11 categories:

- minor distractions (external or other)
- major disruption
- minor technical issues
- major technical issues
- partial lack of understanding the task
- complete lack of understanding the task
- task difficulty
- non-serious attempt
- physical discomfort
- mental discomfort
- no issue

*Minor distractions* were e.g., distracting sounds that occurred for a short time, phone notifications or pop-ups that could be ignored, or when participants simply reported with "yes" on the question whether there were disruptions. *Major disruptions* were e.g., when participants had to leave the screen or reported that they did something else during the attempt.

*Minor technical issues* were one-time instances of necessary reloads or the single occurrence of longer loading times. Responses coded as *major technical issues* were reports that pages had to be reloaded multiple times, or that the server timed out.

A large group of participants expressed a lack of understanding the reasoning task. We distinguished between reports indicating a *complete lack of understanding the task* and a *partial lack of understanding the task*. While a complete lack of understanding was indicated by statements of strong confusion about all tasks, the instructions and training, we concluded a partial lack of understanding when participants stated that some of the items were confusing or if they had difficulties understanding the instructions. The label *task difficulty* was assigned when participants mentioned the challenge of the task without indicating that this was due to their misunderstanding. This includes descriptions of the task itself as being hard as well as attributions of the difficulty to a lack of ability.

The responses of one participant were coded as a *non-serious attempt* as they were merely descriptions of concepts that were being asked, suggesting a computer-generated response. This decision is validated by the fact that this participant omitted responses to all reasoning items.

Some participants reported *physical discomfort* in the form of hurting eyes or headaches. The category *mental discomfort* contains reports of e.g., fatigue, exhaustion, frustration or feeling stressed.

When participants mentioned issues with the last two items of the procedure, that were supposed to be unsolvable to measure persistence (see 2.5.5 on the persistence measure), the variable *Disruptions* was set to *no issue*. *No issue* was also recorded when participants described a positive experience or were thankful for the task. Also, all non-responses were coded as *no issue* in the sense that no issue was reported. This does not exclude unreported disruptions. If participants reported multiple issues, either the most dominant or the most concerning issue was chosen for the categorization.

All open responses and the respective category variable can be found in file https://osf.io/cn6zw (direct document link) in the study's repository at https://osf.io/9j6hm/.

The absolute and relative frequency of the categories are displayed in Table 4. The results show that a majority of participants reported no issues with the study (73.1%). The most often reported issue was the difficulty of the task (10.7%). A fairly large number of participants showed a partial or complete lack of understanding the task (taken together 9.1%). Although some of the reported issues might warrant an exclusion of the data for some analyses, we decided to keep everything in the dataset and add the variable *Disruptions* as a means of a transparent quality description. This way, researchers using the data a) can examine test-taking behavior with the help of the qualitative variable and b) can make their own informed decisions about data exclusion.

### 2.6.5 Quality control of psychometric instruments

After the data collection took place, we assessed the quality of the psychometric instruments used in the study. There are two main criteria according to which we assessed the quality of the instruments. First, we analyzed whether the presumed single-factor model for every instrument fits the data well. For the reasoning test, concentration task, motivational disposition and effort, we fitted congeneric CFA models using the weighted least square mean and variance adjusted (WLSMV) estimator with robust standard errors in lavaan (Rosseel, 2012). For Current Motivation we used maximum likelihood estimation. The scaled versions of the fit indices can be found in Table 5. As the Current Motivation questionnaire and the concentration tasks consist of just three items (for concentration, three items per measurement point), the model was saturated and thus, model fit was perfect. The results show good fit of CFA models to the data for all scales, with the exception of the effort scale failing to meet the RMSEA criterion for good fit.

Secondly, we estimated the reliability of the instruments using the omega coefficient (McDonald, 1999). The results (see Table 6) show that most scales have high or very high reliability, with the exception of the concentration task and the effort scale at the first measurement point. Although the reliability coefficients for these scales are lower, they are not yet as low as to warrant their exclusion from the dataset.

| *DISRUPTIONS* CATEGORY | FREQUENCY |
|---|---|
| minor distractions (external or other) | 12 (0.97%) |
| major disruption | 4 (0.32%) |
| minor technical issues | 8 (0.64%) |
| major technical issues | 4 (0.32%) |
| partial lack of understanding the task | 82 (6.59%) |
| complete lack of understanding the task | 31 (2.49%) |
| task difficulty | 133 (10.70%) |
| non-serious attempt | 1 (0.08%) |
| physical discomfort | 13 (1.05%) |
| mental discomfort | 47 (3.78%) |
| no issue | 909 (73.10%) |

**Table 4** Absolute frequencies of the categories assigned to the nominal variable Disruptions. Percentages of total *N* = 1244 in brackets. See text for information on the category assignment.

| INSTRUMENT | | ω |
|---|---|---|
| Matrix reasoning | 1st measure | .98 |
| | 2nd measure | .97 |
| Concentration | 1st measure | .37 |
| | 2nd measure | .39 |
| | 3rd measure | .43 |
| Motivational disposition | | .87 |
| Effort | 1st measure | .65 |
| | 2nd measure | .74 |
| Current motivation | 1st measure | .85 |
| | 2nd measure | .88 |
| | 3rd measure | .91 |

**Table 6** Omega reliability coefficients of the instruments used in the present study.

| INSTRUMENT | | $\chi^2$(df); *p* | CFI | TLI | 90% RMSEA CONFIDENCE INTERVAL (*p*) | SRMR |
|---|---|---|---|---|---|---|
| Matrix reasoning | | | | | | |
| | 1st measure | 69.24(35); <.001 | >.99 | >.99 | .018 – .038 (>.999) | .020 |
| | 2nd measure | 68.28(45); <.001 | >.99 | >.99 | .018 – .038 (>.999) | .023 |
| Motivational Disposition | | 113.11 (54); <.001 | .98 | .98 | .027 – .041 (>.999) | .044 |
| Effort | | | | | | |
| | 1st measure | 44.25 (5); <.001 | .98 | .96 | .059–0.102 (.01) | .043 |
| | 2nd measure | 51.51 (5); <.001 | .99 | .98 | .066–.109 (.002) | .039 |

**Table 5** Fit indices of the single-factor CFA models for the scales used in this study.

## 2.7 DATA ANONYMISATION AND ETHICAL ISSUES

No information was collected to personally identify the participants. Recruitment and payment were done via Prolific. Anonymous participant IDs are generated by Prolific to identify and discriminate valid data submissions. On the online survey platform SoSciSurvey that we used to collect the data, no other information than the anonymous ID was collected that could identify the participants.

At the beginning of the study, participants were informed about their right to terminate their participation anytime, the data protection policy, the use of their data and the occurrence of attention checks, and subsequently asked about their consent. We informed participants before starting that the study aim concerns a test for logical reasoning. To avoid self-awareness and reactivity, we informed participants about the aim of studying test-taking behavior and motivation only at the end of the study. In this debriefing, we also informed them that the last two reasoning items were not solvable and that we used those to assess their persistence. After being fully informed about the study aims, participants were again asked for their consent to use their data. Participants who revoked their consent still received the full payment, but their data was removed.

The general study idea was ethically approved by the Ethics Committee of the Psychology Department at the Freie Universität Berlin as part of the grant proposal submission.

## 2.8 EXISTING USE OF DATA

There are no current outputs or publications originating from this dataset, yet. A preregistration for validating the ISAR and LBA-P model with this dataset was published before data collection (Mutak & Much et al., 2023). Publications using this dataset for validation of the two models are planned.

## (3) DATASET DESCRIPTION AND ACCESS

### 3.1 REPOSITORY LOCATION

All files are available on OSF in the project's repository at https://osf.io/9j6hm/. It is identified with the DOI: 10.17605/OSF.IO/9J6HM.

### 3.2 OBJECT/FILE NAME

The names, types and a description of all available files can be found in Table 7.

### 3.3 DATA TYPE

The dataset consists of primary data (raw item responses, action sequences and action times for reasoning items, time on task for concentration task items) and processed data (time on task for reasoning items, item scores/correctness indicators, scale scores, disruptions variable).

### 3.4 FORMAT NAMES AND VERSIONS

We provide the main data file as a .csv file which can be read by most statistical software packages. Furthermore,

| FILE NAME | FILE DESCRIPTION |
| --- | --- |
| Main data files | |
| tte_data.csv | Main data file, includes primary data (raw item responses, time on task for concentration task), processed data (time on task for reasoning items, item and scale scores, disruptions variable) |
| tte_data.Rds | Main data file with R operability (data structures, variable labels) |
| tte_codebook.csv | Machine-readable codebook |
| tte_codebook.pdf | Human-readable version of the codebook |
| Supplemental files | |
| ct_ItemOverview.pdf | Depiction of all concentration task items |
| ct_pagetimes.csv | Data file containing primary data: time on page for concentration task items |
| efcm_ItemOverview.pdf | Rephrased items of the Effort and Current Motivation scale, including instructions |
| mr_ItemOverview.pdf | Description of reasoning items including item ID, item figures, solutions and construction rules |
| mr_itemsolutions.csv | Machine-readable file containing reasoning item solution codes |
| mr_logdata.csv | Data file containing primary data: raw action sequences and time measurements for reasoning items |
| tte_disruptions.csv | Data file containing primary data and their interpretation: raw responses to open format questions and the nominal variable subsuming these responses |
| tte_suppl_codebook.pdf | Human-readable codebook for supplemental data (open responses and disruptions, reasoning item solution codes, reasoning log data, and concentration task page time) |

**Table 7** Names and descriptions of the files in the repository at https://osf.io/9j6hm. Files are separated into the folders "Main data files" and "Supplemental Files".

we also provide the data as a .Rds file which includes more functionality (e.g., labels, data structures), but requires R to read it. Supplemental files are provided as .csv files if they contain machine-readable data or as PDF-A files if they are meant to be more accessible.

### 3.5 LANGUAGE

All text in the data collection process and participant communications were in American English. Also, all supplementary material (codebook, preregistration, material descriptions) are given in American English.

### 3.6 LICENSE

The data is published under the license CC-BY 4.0.

### 3.7 LIMITS TO SHARING

There is no identifying information in the dataset and all the Prolific IDs have been replaced by simple numeric IDs generated from our side. Therefore, there are no barriers to the full sharing of the dataset. The data is also not currently under embargo.

### 3.8 PUBLICATION DATE

The data was published in the repository on 28/06/2024.

### 3.9 FAIR DATA/CODEBOOK

In addition to the data, we provide the codebook, which contains the variable names, descriptions, variable types and value ranges (when applicable) for the variables in the published dataset.

In the publication of the dataset, we made sure to conform to the FAIR guidelines. The dataset and supplemental files are findable and accessible via the unique and persistent identifier (DOI) of the repository, the open CC-BY-4.0 license and the metadata attributes attached to the repository and the dataset file. Interoperability and reusability are ensured by publishing the dataset and supplemental data files in a platform-independent machine-readable text format (csv) with accompanying human-readable description files in the platform-independent pdf format. We made sure to apply consistent conventions for naming variables and include the description in the repository.

# (4) REUSE POTENTIAL

### 4.1 STRENGTHS AND LIMITATIONS

The dataset has different strengths that may be useful for use in other settings. First, its experimental design, in which speededness is manipulated, allows for validation of a variety of psychometric models on test-taking behavior. Second, the dataset offers a number of additional variables related to test-taking behavior, which are repeatedly measured and include both self-report and behavioral data. This allows for a comprehensive validation of model parameters that relate to test-taking behavior. Third, the dataset also contains log data, including actions and their respective timing, on the response selection process for the reasoning items. This opens up great possibilities to the development and evaluation of models on more detailed process data, for which hardly any experimental data of this size exist, so far.

The dataset also has its limitations. First, as the participants were recruited via Prolific, where persons take part in studies for money, the results using the data may not necessarily generalize to a different population. Persons in the sample may systematically differ from the general English-speaking adult public in many aspects, including age, education, or motivation to take part in a study. We assessed sociodemographic data and motivation in order to describe the specific sample. Second, the matrix reasoning test seems to have been rather difficult for the Prolific sample. A large number of participants could not solve any of the reasoning test items, resulting in a low test-targeting and reduced variance on the item responses. This may negatively impact the efficiency of person parameter estimation. Third, the study is quite long with a median total time of 50.5 minutes. This could have produced fatigue effects or motivational decline for some participants. The repeated measures of concentration, motivation and effort, or the self-report of mental discomfort in the disruptions variable may describe some of these effects. Fourth, while there are quality controls to ensure the collection of valid data, we cannot fully rule out issues arising with the data collection outside of a controlled lab-setting. Especially the timing data provided in the dataset may not exclusively represent time used for solving a task but may also reflect breaks or interruptions. By allowing for explicit breaks during the study and assessing interruptions, we aimed to minimize this or assess it. Timing data should, thus, be carefully examined before its use and extreme response times should be used with caution. One possibility is to remove outliers on response times or to investigate their impact on the results in sensitivity analyses.

### 4.2 POTENTIAL USE CASES

The data is well suited for the development and validation of psychometric models on test-taking behavior, including models on test-engagement, item omissions, random and informed guessing, speed-accuracy trade-off, and response strategies. The dataset provides data not only on responses and response times, but also on confidence ratings, action sequences as well as their timing data. This allows for the application of a variety of psychometric modeling strategies. The experimental design and the extensive assessment of confounder variables allows a thorough evaluation and validation of model parameters.

Apart from psychometric modeling, the data is useful for the general investigation of test-taking behavior in

a low-stakes achievement test without time-limits. The data can offer insights into the evolution of fatigue and motivation over the course of a testing session and its relation to performance. It can also offer insights into the effects of speededness instructions on persistence, performance, and motivation and provides data for the investigation of the different relationships of speed and ability, e.g., speed-ability trade-offs. As action sequences and respective timing data are provided, the dataset offers a unique opportunity to investigate the underlying solution process for correct and incorrect responses. The dataset also includes confidence ratings during the training that allow for a metacognitive context in investigating test-taking behavior. In the reasoning test, there are rarely occurring item omissions (ranging from 0.2% to 3.7% per item) which allow for a limited investigation of processes underlying item omissions.

## NOTES

1   The instruction for the speeded condition was: "Please solve the items in this block **as fast as possible** while still being accurate. It's okay if you're not sure about your response.

    You will be reminded of the focus on **speed** at the top of each page." The non-speeded instruction read: "Please solve the items in this block **as accurate as possible**. If you're not sure about your response, you may take your time to find a solution. You will be reminded of the focus on **accuracy** at the top of each page." The text was displayed in a highlighted text box with the shown bold face markings and font colors for the words *speed* and *accuracy*.

2   Due to how Prolific communicates with the study platforms, it is impossible to record a person's participation unless they reach the last page of the survey.

3   Our budget was planned with an estimated duration of the study of 50 minutes and the given costs on Prolific at the time of planning. This resulted in 1000 participants. However, due to offers on Prolific and deviations from the estimated time needed, the actual number of participants deviated from the planned one. We collected as much data as our budget did allow for.

4   The unequal assignment to groups was due to estimation requirements of the different models. See the preregistration of the study (Mutak & Much et al., 2023) for details.

## ACKNOWLEDGEMENTS

We are grateful to our student assistants Karoline Langner, Niklas Neek, and Hanna Reda for their contributions to the preparation of this study. We like to thank all colleagues and friends who gave feedback during the study development.

## FUNDING INFORMATION

The study was funded by the Deutsche Forschungsgemeinschaft (DFG, German Research Foundation) within the project "Test-taking engagement and test-taking behavior: Modeling the processes underlying item nonresponse and guessing" (DFG project number 28872689, grants RA 3453/1-2 and PO 1655/3-2).

## COMPETING INTERESTS

The authors have no competing interests to declare.

## AUTHOR CONTRIBUTIONS

Sören Much and Augustin Mutak are shared first authorship.
    Conceptualization: S.M., A.M., S.P., and J.R.
    Data curation: S.M. and A.M.
    Methodology: S.M., A.M., S.P., and J.R.
    Writing – original draft: S.M. and A.M.
    Writing – review & editing: S.M., A.M., S.P., and J.R.

## AUTHOR AFFILIATIONS

**Sören Much** (ID) orcid.org/0000-0002-9837-8073
Martin-Luther-Universität Halle-Wittenberg, DE; Freie Universität Berlin, DE
**Augustin Mutak** (ID) orcid.org/0000-0001-9003-1166
Freie Universität Berlin, DE
**Steffi Pohl** (ID) orcid.org/0000-0002-5178-8171
Freie Universität Berlin, DE
**Jochen Ranger** (ID) orcid.org/0000-0001-5110-1213
Martin-Luther-Universität Halle-Wittenberg, DE

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

## PEER REVIEW COMMENTS

*Journal of Open Psychology Data* has blind peer review, which is unblinded upon article acceptance. The editorial history of this article can be downloaded here:

- **PR File 1.** Peer Review History. DOI: https://doi.org/10.5334/jopd.124.pr1

**TO CITE THIS ARTICLE:**

**Submitted:** 13 September 2024     **Accepted:** 11 March 2025     **Published:** 21 March 2025

