## [Peer Review History. · Journal of Open Psychology Data]

Peer Review Comments: Data from a validation study of two psychometric models on test-taking behavior

Dear Sören Much, Augustin Mutak, Steffi Pohl, Jochen Ranger,

After review, we have reached a decision regarding your submission to Journal of Open Psychology Data, "Data from a validation study of two psychometric models on test-taking behavior". Our decision is to request revisions of the manuscript prior to acceptance for publication. In sum, the revisions should provide greater clarity in procedure and processes, and further consistency in presentation of your work, and I hope with the clear guidance offered this shouldn't take too much more of your time.

The full review information is included at the bottom of this email. Please note that there may also be a copy of the manuscript file with reviewer comments available once you have accessed the submission account. In addition to their comments, please do thoroughly work your way through the manuscript to ensure you are being as clear and precise to ensure your work is as accessible and unambiguous as possible.

Instructions for how to resubmit your article online are pasted below. Please ensure that your revised files adhere to our author guidelines, and that the files are fully proofed prior to upload. Please also include a revised version of your article with 'tracked changes', adding comments where appropriate, to indicate the revisions made, in addition to a brief document outlining how you have responded to the reviewers' requests.

If you have trouble processing the revisions, our Help Center (<https://help.u-community.io>) or downloadable PDF (<https://bit.ly/Author-Guide-OJS-3>) may be able to help. If not, please get in touch and we'll be happy to help.

Please also ensure that all copyright permissions have been attained for any figures/tables you have included.

Please could you have the revisions submitted with two weeks. If you cannot make this deadline, please let us know as early as possible.

Kind regards,

Prof Thomas Rhys Evans

Reviewer A:

Recommendation: Revisions Required

Comments to the author(s)

2.4.2. Sample description - change to self-identifying gender language. Give details of other gender if you have it.

SD - change to 2 decimal places
and average of 28 years instead of 28.33

2.6.2 Pretest of the procedure - any demographic information on the 17 participants?

Procedure section should be included - there is information of the procedure in the design but think it would be clearer if there was a procedure section.

Statistical letters (e.g., *N*) should be in italics

The rest looks great - links can be opened and I enjoyed reviewing.

Reviewer P:
Recommendation: Accept Submission

Comments to the author(s)

Thank you for the opportunity to review this data paper. The study is well-designed. Overall, I found the description clear, and the dataset is well-documented and easily accessible. I have no major suggestions for the authors, but I would like to highlight a few minor points:

- I suggest reporting the mean task duration in the main text, preferably towards the end of the "Study Design" section.

- You mention that "participants were not aware that test-taking behavior is being investigated." How was the study presented on Prolific? What did participants believe the study's purpose was?

- I recommend revising the language to improve clarity and reduce repetition, particularly in the Background section. Additionally, there are some punctuation errors. For example, in the sentence "A linear latent growth term is specified for both, ability and speed...", the comma after "both" should be removed. Also, the following sentence needs revision: "To be included in the study, participants had to [...] are not in the process of being diagnosed and do not strongly suspect to have undiagnosed dyslexia."

- In Figure 2, could you add the correct solution to the item? It would help readers better understand the task... and I'm also curious to see if my answer was correct!